

# Brazilin cream from *Caesalpinia sappan* inhibit periodontal disease: *in vivo* study

Vinna Kurniawati Sugiaman[1], Jeffrey Jeffrey[2], Silvia Naliani[3], Natallia Pranata[1], Shelly Lelyana[4], Wahyu Widowati[5], Rival Ferdiansyah[6], Dhanar Septyawan Hadiprasetyo[7,8] and Vini Ayuni[8]

[1] Department of Oral Biology/Faculty of Dentistry, Maranatha Christian University, Bandung, West Java, Indonesia

[2] Department of Pediatric Dentistry/Faculty of Dentistry, Universitas Jenderal Achmad Yani, Cimahi, West Java, Indonesia

[3] Department of Prosthodontics/Faculty of Dentistry, Maranatha Christian University, Bandung, West Java, Indonesia

[4] Department of Oral Medicine/Faculty of Dentistry, Maranatha Christian University, Bandung, West Java, Indonesia

[5] Faculty of Medicine, Maranatha Christian University, Bandung, West Java, Indonesia

[6] Department of Pharmaceutics, Sekolah Tinggi Farmasi Indonesia, Bandung, West Java, Indonesia

[7] Faculty of Pharmacy, Universitas Jenderal Achmad Yani, Cimahi, West Java, Indonesia

[8] Biomolecular and Biomedical Research Center, Aretha Medika Utama, Bandung, West Java, Indonesia

Corresponding author
Vinna Kurniawati Sugiaman, vinnakurniawati@yahoo.co.id

## ABSTRACT

**Background**. Gingivitis is an inflammation of the gums that is the initial cause of the development of periodontal disease by the activity of Nuclear Factor-kappa B (NF-κB), Interleukin-1β (IL-1β), Interleukin-6 (IL-6), p38, and Tumor Necrosis Factor-α (TNF-α). Unaddressed chronic inflammation can lead to persistent disturbances in other parts of the body. Brazilin is a naturally occurring plant chemical that may have antibacterial and anti-inflammatory effects. Treatment based on the natural plant compound, brazilin, is developed in the form of a topical cream for easy application.

**Objective**. The aim is to develop the natural compound brazilin in the form of a topical cream as an anti-inflammatory agent to reduce NF-κB expression through Imunohistochemistry (IHC) methods, and the expression of pro-inflammatory genes IL-1β, IL-6, p38, and TNF-α.

**Methods**. Male Sprague-Dawley rats were induced with gingivitis using *P. gingivalis* bacteria. The observed groups included rats treated with a single application of brazilin cream and rats treated with two applications of brazilin cream. The treatment was administered for 15 days. On days 3, 6, 9, 12, and 15, anatomical wound observations and wound histology using hematoxylin-eosin and Masson's Trichrome staining were performed. NF-κB protein expression was analyzed using the IHC method. Gingival inflammation gene expression of NF-κB, IL-1β, IL-6, p38, and TNF-α was measured using q-RTPCR.

**Results**. Single and double applications of brazilin cream increased angiogenesis and decreased NF-κB protein expression, in addition to the IL-1β, IL-6, p38, and TNF-α gene expressions.

**Conclusion**. In a rat gingivitis model, Brazilin cream may function as an anti-inflammatory agent in the gingival tissue.

## INTRODUCTION

Periodontal disease (PD) is a condition characterized by inflammatory wounds with signs of damage to the tissues supporting the teeth, including the gingival tissue and connective tissue that attaches the teeth to the alveolar bone (periodontal ligament) (*Paul et al., 2021*). A bad microbiome situation close to the gum line, called periodontal disease (*Abusleme et al., 2013*), is marked by differences in the microbes that live there and how the body's immune system reacts to them. One of these disorders, gingivitis, is characterized by gingival inflammation due to the cumulative influence of certain microbes. Gingivitis can develop into periodontitis, a condition marked by the deterioration of tissues supporting the teeth, if treatment is not received (*Löe et al., 1986*). Science acknowledges gingivitis as the precursor of several periodontal illnesses, including periodontitis, even though some cases of gingivitis are benign and treatable. Gingivitis is gum inflammation caused by a buildup of dental plaque, characterized by redness and swelling of the gums without loss of tooth attachment. If left uncontrolled, gingivitis can progress to more serious periodontitis. Good control of gingivitis can prevent periodontal tissue damage (*Trombelli et al., 2018*). The World Health Organization's records indicate that periodontitis affects approximately 35–50% of people worldwide, with gingivitis affecting 9–17%. During puberty, this number rises to 70–90%, accounting for around 47% of the adult population (*Holm-Pedersen, Walls & Ship, 2015*; *Eke et al., 2012*). Other prevalences of gingivitis in the records (*Fan et al., 2021*) ranged from 23 to 77% in young Latin American individuals and 28.58% in children aged 6–12 years in Jinzhou, China.

Several factors contribute to periodontal disease, including excessive and unrestricted smoking, poor dental hygiene, unrecommended drug consumption, advanced age, and disturbed mental conditions (stress) (*Nazir, 2017*). Risk factors for poor oral hygiene cause more gingival crevicular fluid (GCF) to flow because more pathogen-associated molecular patterns (PAMPs) cause bacterial plaque to form (*Cekici et al., 2014*). Inflammation of cells close to the periodontal epithelium is caused by the growth of gram-negative bacteria, which sets off the innate immune pathway (*Fujita et al., 2018*; *Kumar, 2017*). We know that toll-like receptors (TLRs) cause inflammation and that they also cause inflammation in the periodontal ligament, gingival fibroblasts, and dendritic cells (*Song et al., 2017*; *Behm et al., 2019*). Some cytokines and chemokines that cause inflammation are Tumor Necrosis Factor-α (TNF-α), Interleukin-1β (IL-1β), Interleukin 6 (IL-6), Interleukin 8 (IL-8), Interleukin 12 (IL-12), Interleukin 17 (IL-17), and receptor activator of nuclear factor kappa β ligand (RANK-L). These chemicals are more commonly found in cells that are close to each other in connective tissue and alveolar bone (*Duka et al., 2019*).

More than 70,000 people took part in the study by *Da Silva et al. (2017)*, who looked at genetic variations in genes like IL-1α, IL-1β, IL-6, IL-10, and matrix metalloproteinase (MMP-3). These variations were carefully looked over and compared to see how much the genome was affected by periodontal genetic factors. The results showed that MMP-9

is significantly linked to the risk of getting periodontitis. Also, nuclear factor kappa B (NF-κB) is important for the swelling and irritation that affects the gums and alveolar bone around teeth, especially when gum diseases like gingivitis and periodontitis are present (*Ghafouri-Fard et al., 2022*). In certain conditions, elevated NF-κB expression can exacerbate inflammation (*Arabaci et al., 2014*).

Southeast Asia is home to the plant *Caesalpinia sappan* L., also known as brazilian wood or sappan wood, which belongs to the leguminous plant family. Natural compounds such as brazilin, found in sappan wood, have resulted to its widespread use as food or a traditional medicinal drink (*Toegel et al., 2012*). Studies have discovered the potential of the substance known as brazilin to protect cells from oxidative stress, along with its anti-inflammatory, antibacterial, hypoglycemic qualities, and its biological effects on a variety of pathologies such as fibrillogenesis, osteoarthritis, Parkinson's disease, Alzheimer's disease, and others (*Nava-Tapia et al., 2022*).

Topical creams and gels enable the targeted delivery of active ingredients to specific areas, reducing the risk of impacting beneficial bacteria in the oral microbiome (*Amiri et al., 2021*). The pharmaceutical industry has extensively explored the use of cream-based applications on gums due to the biodegradable and biocompatible properties of these creams. People widely use cream because it improves patient tolerance, has fewer side effects, tends to be non-toxic, rarely causes allergies, doesn't irritate the eyes or skin, and has lower production costs (*Deogade, Deshmukh & Sakarkar, 2012*). Calendula and aloe vera, two anti-inflammatory creams and gels, have demonstrated their ability to reduce inflammation and promote healing. Therefore, researchers came up with a new way to use the active compound brazilin from *C. sappan* plants. They made an ointment cream and tested it on an animal model of gingivitis by looking at gene expression of NF-κB, IL-1β, IL-6, p38, and TNF-α. The results showed that the cream could help prevent gingivitis, which leads to periodontal disease.

## MATERIALS & METHODS

### Preparation of brazilin cream sample

Brazilin isolation was obtained from Sappan (*Caesalpinia sappan* L.) wood powder through a modified maceration process (*Widowati et al., 2019*). Sappan wood was extracted using 96% ethanol for 24 h, which was soaked three times. While the extraction process took place twice, the macerate was separated from the residue, which is then evaporated at room temperature (25 °C). The macerete was washed with hot water and then dried. The crude was added with methylene chloride 1:1, which was then stirred and filtered. The residue that was separated from the filtrate was added to chloroform 1:2, which is then used as filtrate after being separated from the residue. The filtrate was evaporated, and the isolate was obtained. In the process of the separation and purification of brazilin, fractions from sappan wood extract were isolated using vacuum column chromatography and gravity column chromatography. The separated fractions were then further purified using preparative thin-layer chromatography (KLT) (*Sari, Widiyantoro & Alimuddin, 2018*). The resulting isolates were then tested for purity using one-dimensional and two-dimensional

KLT before elucidating the structure of brazilin using instrument analysis techniques such as NMR-1H (*Kim et al., 2015*).

The brazilin isolate preparation that was owned will be used to make a cream that has been modified from *Ahmad et al. (2019)*. The production of the cream was carried out in a two-stage process. Stage one was dissolving the formulation into 70 grams of pure coconut oil (VCO), assisted by using a sonicator for 60 min at a maximum temperature of 60 °C. The second stage was to dissolve the surfactant (sucrose monoester 1750) and cosurfactant (glycerol) into a beaker and stir until cream was formed. Finally, the first and second stages were mixed little by little while continuing to stir on a hot plate for 30 to 40 min (*Widowati et al., 2023*; *Ahmad et al., 2019*).

## Induction of gingivitis in rats

The experiment was conducted at the iRATco Animal Laboratory Services animal facility in Bogor, Indonesia, with five animals in each group. All testing protocols described have been approved by the Animal Ethics Committee of iRATco VLS with number 4.2.012-3/KEHI//I/2023. Gingivitis induction was performed following the notes (*Harris, 2023*) with slight modifications. This research was carried out on male Sprague Dawley rats obtained from the animal facility, aged 18–20 weeks with a body weight of around 150–200 g. The animals selected were those that met the inclusion criteria (male, Sprague Dawley breed, weighing 150–200 g, and 18–20 weeks old), and the non-inclusion criteria from this study were rats that were deformed and diseased prior to testing. Animals were allowed to acclimatize for a week in an Individually Ventilated Cage (IVC) measuring 30.5 × 20.5 × 15.5 cm and with a 2 cm high sawdust mattress that is replaced every 2–3 days (*Hidayat et al., 2022*). The temperature provided was individually controlled and ventilated at 22 °C and 40% relative humidity with a 12-hour light-dark cycle. Rats were given free access to food containing 5% crude fiber, 18% crude protein, and 50% crude fat (PT Indoofeed) and water ad libitum for acclimatization (*Kurniati, Garmana & Sakinah, 2021*; *Widowati et al., 2022*). There are six groups in this study, which are romanized as I, II, III, IV, V, and VI in the table and figure. The description includes I; a negative control group (normal rats); II; a positive control group (bacteria-induced rats); III; a vehicle control group (single oral base cream); IV; drug control (single patent drug); V; test group 1 (single Brazilin cream); and VI; test group 2 (double Brazilin cream). The drug used in the drug group is triacinolone acetonide ointment 0.1% (DKL1124400130A1; Taisho) (*Kumar et al., 2023*). The test, drug, and positive groups will be given local anesthesia to induce gingivitis using P. gingivalis bacteria at a concentration of $2 \times 10^8$ per rat. The volume of bacteria injected was 50 µl with a ratio of 1:1. Bacteria are injected into the gingiva using a Nichimate stepper. The inducer was applied every 3 days for 14 days or until gingivitis occurred in all samples. All groups received the same housing and feeding regimen. The treatment of animals in this study group (I, II, III, IV, V, VI) implemented a protocol to induce experimental periodontitis using incisor ligatures and gingival tissue taking. Taking gingival tissue is intended to determine the extent of inflammation caused by bacteria and the effectiveness of brazilin cream in reducing this macroscopically and microscopically at the next stage. Gum inflammation, swelling, and unnatural redness are signs of gingivitis. After 14 days

of induction, all groups of rats were euthanized using isoflurane overdose *via* inhalation (*Elhaieg et al., 2023*). Then the rat's jaw was removed and soaked in 10% formalin for two days. The rat jaws were then soaked again in a weak acid solution of formic acid (EMSURE, 100264) for a decalcification process for 10 days before analysis. Rats that were sick, die, or did not gain weight during acclimatization were executed. At the end of the probationary period, those who were still alive were executed.

## Wound histopathological examination

The Masson Trichome (MT) staining procedure entails immersing the tissue of all test groups of rats in Weigert iron hematoxylin for 10 min, followed by rinsing with distilled water and staining with Bieblich Scarlet Acid Fucin solution for 10–15 min. After another rinse with distilled water, the slides were stained for 10–15 min using a phosphomolybdenum-phosphotungstic acid solution. After being dyed for between 5 and 10 min with an aniline blue solution, the tissues were submerged in a 1% acetic acid and 95% absolute ethanol solution. The examination was carried out under a microscope covered with glass. The parameters measured were the area of inflammation, angiogenesis, and collagen tissue area using a polarized light microscope (Leica, Wetzlar, Germany) (*Laksmitawati et al., 2023*; *Suvik & Effendy, 2012*).

## Immunohistochemistry assay (IHC) for NF-κB expression

The gingival tissue of all test groups of rats to be fixed began with preservation in paraffin. Deparaffinization was carried out for 15 min at a temperature of 56 °C, and then the glass object was rinsed with xylene. Once the paraffin was removed, the sample was rehydrated using different ethanols (absolute, 90%, and 70%), and soaked for an extra 30 min after being rinsed with phosphate buffered saline (PBS). At a temperature of 121 °C for 10 min, antigen retrieval was carried out in citrate buffer (pH 6; Abcam ab208572). To block endogenous peroxidase, samples were treated in methanol (Merck, 106009) with 3% $H_2O_2$ (Merck, 107209). After incubating with 5% bovine serum albumin for 10 min, we carried out primary antibody reactions overnight at room temperature. Visualization of target proteins was using the HRP/DAB detection rabbit-specific IHC Kit (ABC) (Abcam, ab64261). After making a hematoxylin counterstain, the stained tissue was examined under a Primostar microscope (Zeiss). A photograph was obtained by Lumenera Infinity 1-3c. A qualitative comparison of IHC results was based on the intensity of expression to be depicted in graphical form. ImageJ software was utilized for quantification of the positive index on immunohistochemistry slides (*Rosni et al., 2021*; *Widowati et al., 2022*).

## Quantification of gingival tissue TNF-α, IL-1β, IL-6, and p38 by q-RTPCR

Total rat gingival RNA tissues from all test groups of rats were extracted and purified using the Direct-zol RNA Miniprep Plus Kit (R2073; Zymo) in accordance with the manufacturer's instructions. Utilizing iScript Reverse Transcription Supermix for RT-PCR (170-8841; Bio-Rad, Hercules, CA, USA), complementary DNA synthesis was performed. We used the Agilent AriaMx 3000 real-time PCR technology to evaluate gene expression quantitatively. Evagreen Master Mix (1725200; Bio-Rad, Hercules, CA, USA) was the qPCR

**Table 1  Primer sequences used in qRT-PCR.**

| Gene | Forward (5′–3′) | Reverse (5′–3′) | Product Length (bp) | Annealing temperature (C) | Reference |
|---|---|---|---|---|---|
| IL-1β | AGAATACCACTTGTTGGCT | GTGTGATGTTCCCATTAGAC | 134 | 55 | NM_031512.2 |
| IL-6 | GAGCATTGGAAGTTGGGGTA | TGATGGATGCTTCCAAACTG | 230 | 53 | NM_012589.2 |
| TNF-α | GAAGACAATAACTGCACCCA | AACCCAAGTAACCCTTAAAGTC | 138 | 54 | NM_012675.3 |
| P-38 | AGATAATGCGTCTGACGGG | AGGGGATTGGCACCAATAAA | 139 | 58 | NM_031020.3 |

**Table 2  Purity of gingival RNA treated with brazilin cream.**

| No. | Sample | Concentration (ng/μl) | Purity (λ260/λ280 nm) |
|---|---|---|---|
| 1 | I | 47.8 | 2.211 |
| 2 | II | 54.875 | 2.204 |
| 3 | III | 60.65 | 2.441 |
| 4 | IV | 16.225 | 2.045 |
| 5 | V | 90.525 | 2.203 |
| 6 | VI | 41.375 | 2.219 |

**Notes.**

I: Negative control (normal rat), II: Positive control (bacterial induction rat), III: Vehicle control (1 x oral base cream), IV: Drug control (1 x patent medicine), V: II + 1 x brazilin cream, VI: II + 2 x brazilin cream.

reaction mix used (*Widowati et al., 2020*; *Widowati et al., 2021*). Table 1 displays the primer sequences (Macrogen), and the Table 2 displays the RNA's concentration and purity, which were measured spectrophotometrically at 260/280 nm.

## Statistical analysis

Statistical analysis was carried out with the help of SPSS 22.0 software. One-way analysis of variance (ANOVA) and *post-hoc* Tukey dam value *($p < 0.05$) to be considered significant.

## RESULTS

### Macrogingiva

We check the gingival tissue of rats given induced gingivitis after 15 days to qualitatively assess the effect of the given treatment. Rat gingival tissue (Table 3) using *P. gingivalis* bacterial induction began to appear on day 3, producing a rat model of gingivitis. The resulting signs are the buildup of dental plaque and tooth decay due to tartar until the 15th day. The brazilin cream (V and VI) treatment group demonstrated (Fig. 1) that there was a gradual macro-improvement from day to day in the gingival condition of the gingivitis model rats. This indicates that the treatment with brazilin cream (V and VI) improved the gingiva condition.

### Histopathology of inflammation and angiogenesis of gingival tissue in rat gingival model by HE method

HE staining allows for the histopathological evaluation of inflammation and angiogenesis in rat gingival tissue. According to Fig. 2A, the inflammation score graph shows that groups V and VI experienced a decrease in inflammation and had a significant difference ($p < 0.05$)

**Table 3  Macrogingival results in gingivitis model rats with brazilin cream treatment group.** I: Negative control (normal rat), II: Positive control (bacterial induction rat), III: Vehicle control (1 x oral base cream), IV: Drug control (1 x patent medicine), V: II + 1 x brazilin cream, VI: II + 2 x brazilin cream.

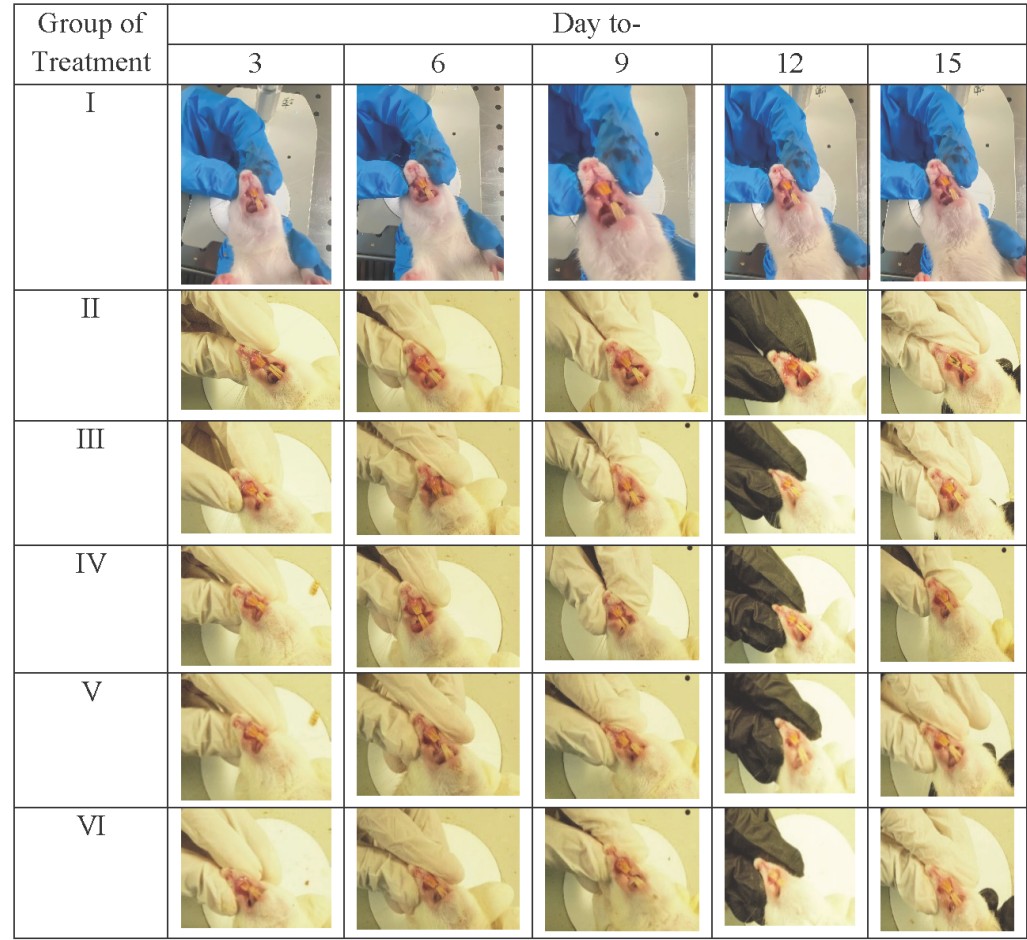

| Group of Treatment | Day to- | | | | |
|---|---|---|---|---|---|
| | 3 | 6 | 9 | 12 | 15 |
| I | | | | | |
| II | | | | | |
| III | | | | | |
| IV | | | | | |
| V | | | | | |
| VI | | | | | |

with the positive control group caused by bacteria. Furthermore, the angiogenesis score in Fig. 2B shows that giving brazilin (groups V and VI) raises the angiogenesis score compared to group II (the positive control). Putting brazilin cream on the rat gingiva increases the number of fibroblast cells, new blood vessels, and the inflammatory process, which helps the rat gingival wound heal as shown in Fig. 3.

## Gingival NF-κB protein expression in gingivitis rat model using IHC methods

Evaluation of NF-κB expression using the IHC method in Fig. 4 shows that there is a significant effect of using brazilin cream on rat gingival tissue. All treatment groups were able to reduce NF-κB expression levels compared to the positive group (group II). The most effective application of brazilin cream in reducing Nf-KB expression levels is two applications of brazilin cream (group VI).

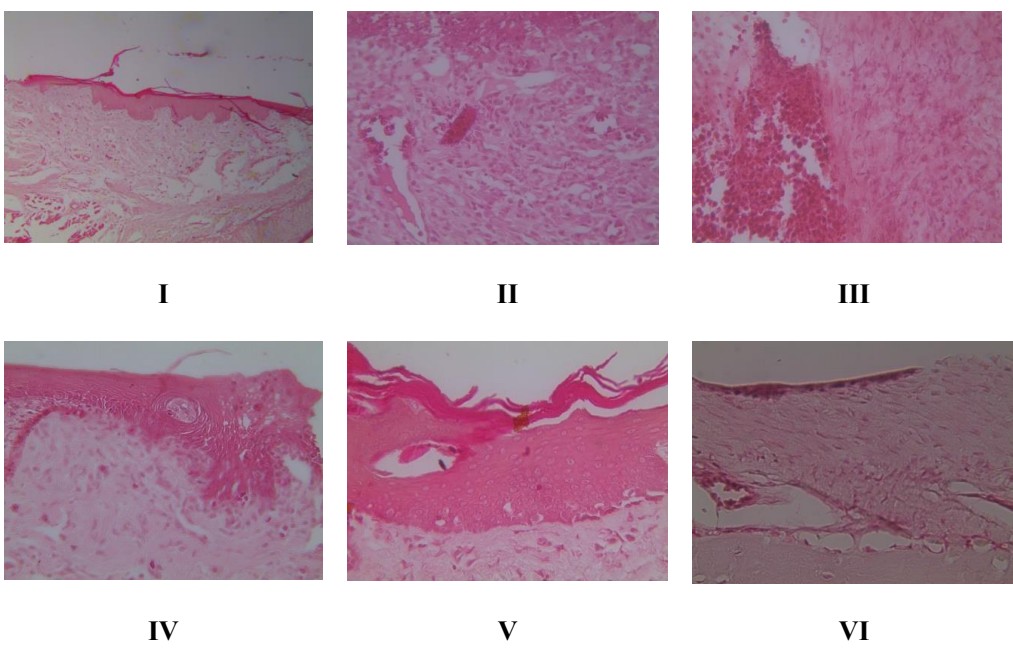

**Figure 1  Effect of brazilin cream toward inflammation and angiogenesis of gingival tissue in gingivitis model rats using HE method (40x magnification).** I: Negative control (normal rat), II: Positive control (bacterial-induced rat), III: Vehicle control (1 x oral base cream), IV: Drug control (1 x patent medicine), V: II + 1 x brazilin cream, VI: II + 2 x brazilin cream.

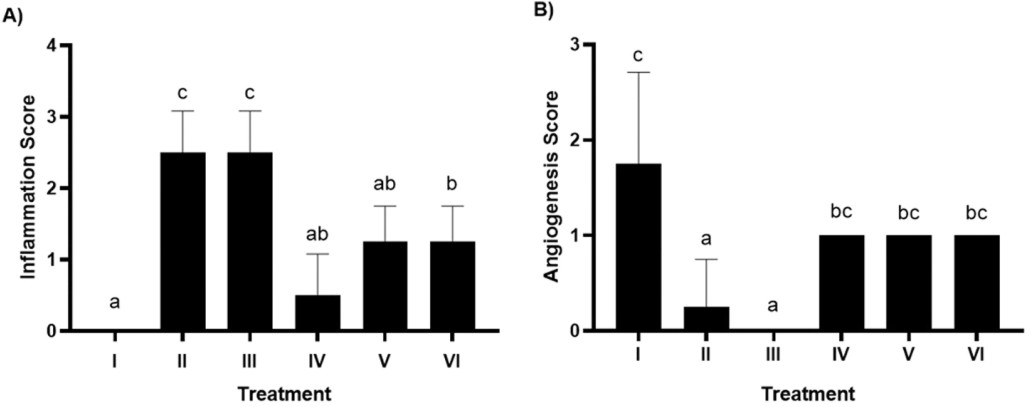

**Figure 2  Effect brazilin cream toward histopathology of gingival tissue gingivitis rats model.** (A) Inflamation. (B) Angiogenesis. Data are presented as mean ± standard deviation. For each treatment, the assay was performed in four repetitions I: Negative control (normal rat), II: Positive control (bacterial induction rat), III: Vehicle control (1 x oral base cream), IV: Drug control (1 x patent medicine), V: II + 1 x brazilin cream, VI: II + 2 x brazilin cream. Different lowercase letters (a, ab, bc, c) for inflammation score and different lowercase letters (a, bc, c) mark significant differences among treatment groups based on Kruskal Wallis and Mann–Whitney tests ($P < 0.05$).

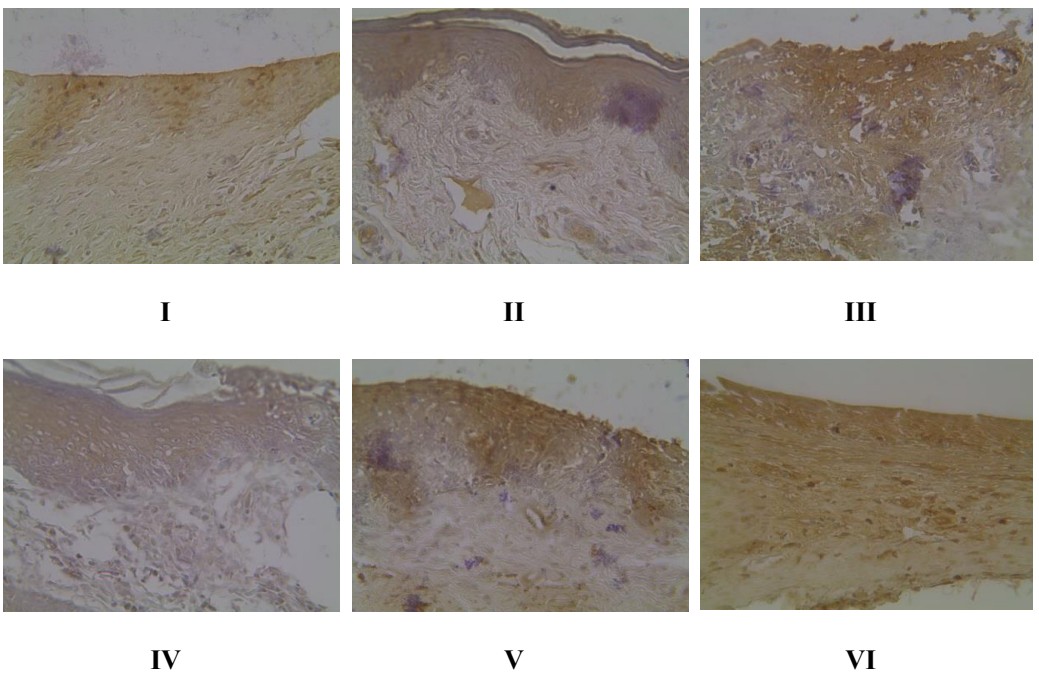

I                     II                     III

IV                     V                     VI

**Figure 3  Effect of brazilin cream toward gingival NF-κB protein expression using IHC methods (40x magnification).** I: Negative control (normal rat), II: Positive control (bacterial induction rat), III: Vehicle control (1 x oral base cream), IV: Drug control (1 x patent medicine), V: II + 1 x brazilin cream, VI: II + 2 x brazilin cream.

## Histopathology of collagen scores in rat gingival models using Masson trichome (MT) method

The Masson trichome (MT) method (Fig. 5) from *Suvik & Effendy (2012)* determined the collagen value. Treatment with 1 and 2 times the application of brazilin cream was able to increase the collagen score, although visually there wasn't much difference. However, the twice-applied brazilin cream group (group VI) was more effective in increasing the collagen score (Fig. 6) (*Suvik & Effendy, 2012*).

## The effect of brazilin cream on IL-1, IL-6, p-38, and TNF-α gene expression

The impact of using brazilin cream on the levels of p-38, TNF-α, IL-1β, and IL-6 gene expression in the gingival tissue of rats suffering from gingivitis is displayed in Fig. 7 (A, B, C,D). In comparison to the positive control group (Group II), the levels of p-38, TNF-α, IL-1β, and IL-6 went down when the brazilin cream was used ($p < 0.05$). Brazilin cream applied twice a day (group VI) is the most effective therapy to reduce TNF-α, p-38, IL-1β, and IL-6 levels.

## DISCUSSION

The main reason for tooth loss is periodontitis, an inflammatory condition brought on by bacterial infections in the mouth (*Chen et al., 2021*). Inflammation caused by a lack

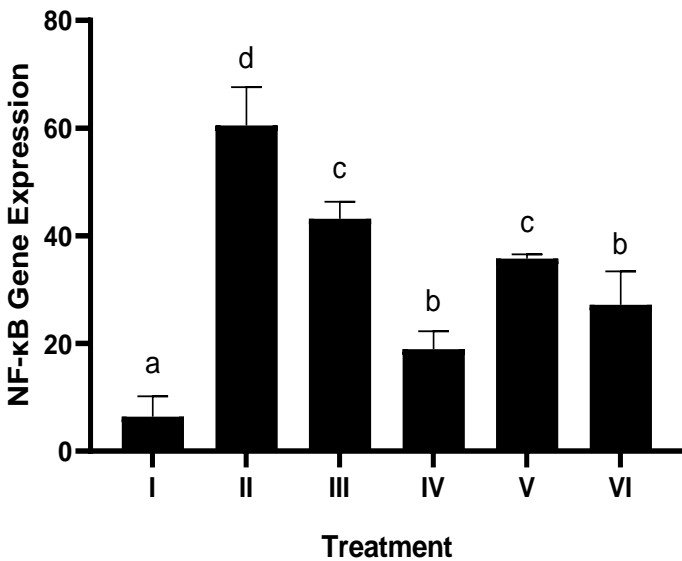

**Figure 4** **Effect of brazilin cream toward gingival NF-κB protein expression in gingivitis rats model.** Data are presented as mean ± standard deviation. For each treatment, the assay was performed in four repetitions. I: Negative control (normal rat), II: Positive control (bacterial induction rat), III: Vehicle control (1 x oral base cream), IV: Drug control (1 x patent medicine), V: II + 1 x brazilin cream, VI: II + 2 x brazilin cream. Different lowercase letters (a, b, c, d, e) mark significant differences between treatment groups ($P < 0.05$, One way ANOVA and Tukey HSD test).

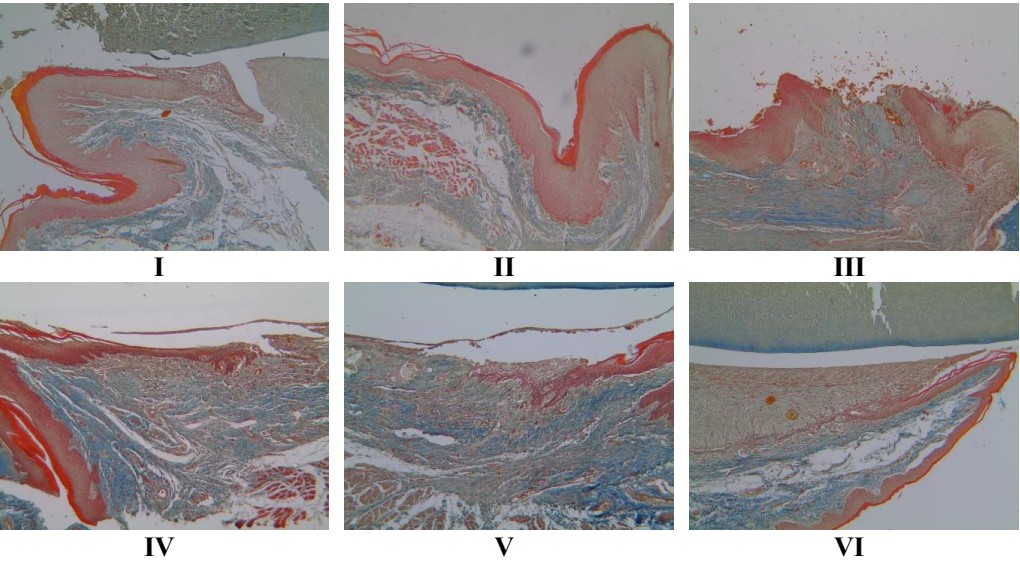

**Figure 5** **Effect of brazilin cream toward collagen scores in rat gingival with MT methods (40x magnification).** I: Negative control (normal rat), II: Positive control (bacterial induction rat), III: Vehicle control (1 x oral base cream), IV: Drug control (1 x patent medicine), V: II + 1 x brazilin cream, VI: II + 2 x brazilin cream.

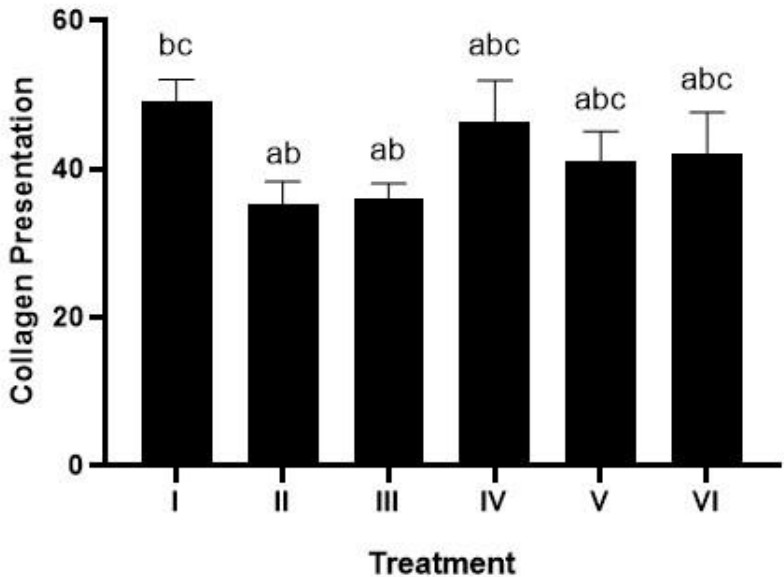

**Figure 6 Effect of brazilin cream toward collagen in gingivitis rats model.** Data are presented as mean ± standard deviation. For each treatment, the assay was performed in four repetitions. I: Negative control (normal rat), II: Positive control (bacterial induction rat), III: Vehicle control (1 x oral base cream), IV: Drug control (1 x patent medicine), V: II + 1 x brazilin cream, VI: II + 2 x brazilin cream. Different lowercase letters (a, ab, abc, bc) mark significant differences between treatment groups ($P < 0.05$, one way ANOVA and Tukey HSD test).

of antioxidants results in an overactive inflammatory response (*Arulselvan et al., 2016*). Antioxidant-containing natural or synthetic ingredients can treat inflammation. In this case, we administered therapy using the natural compound brazilin. In this case, we employed brazilin, a formulated natural compound, as a treatment. We administered brazilin cream as a treatment because it reportedly has antioxidant, anti-inflammatory, and antibacterial properties that reduce proinflammatory gene expression (*Nirmal et al., 2015*). The research evaluation began with an examination of the gingival tissue of rats that had been induced by bacteria. After 15 days of bacteria-induced inflammation in the rats' gingival tissue, the results emerged (Fig. 1). Previous studies reported that plaque bacteria such as *P. gingivalis* caused the swelling and inflammation of the gingival tissue (*Rani et al., 2022*).

Macrogingiva in rats refers to observable outcomes or changes at a gross or macroscopic level in the gingival tissue of rats (Table 3). This entails observing changes that are visible on a larger scale, such as changes in color, swelling, or bleeding in the rat's gums. Macrogingival outcomes in rats often serve as the initial basis for assessing gum health and identifying early signs of periodontal issues.

Our study highlights the application of brazilin cream on rats' gingiva, which shows a decrease in the inflammatory process, the creation of new blood vessels, and a rise in fibroblast cells. This phenomenon proves beneficial in facilitating the healing process of gingival wounds in rats. Research by *Baru et al. (2021)* aligns with this result, suggesting

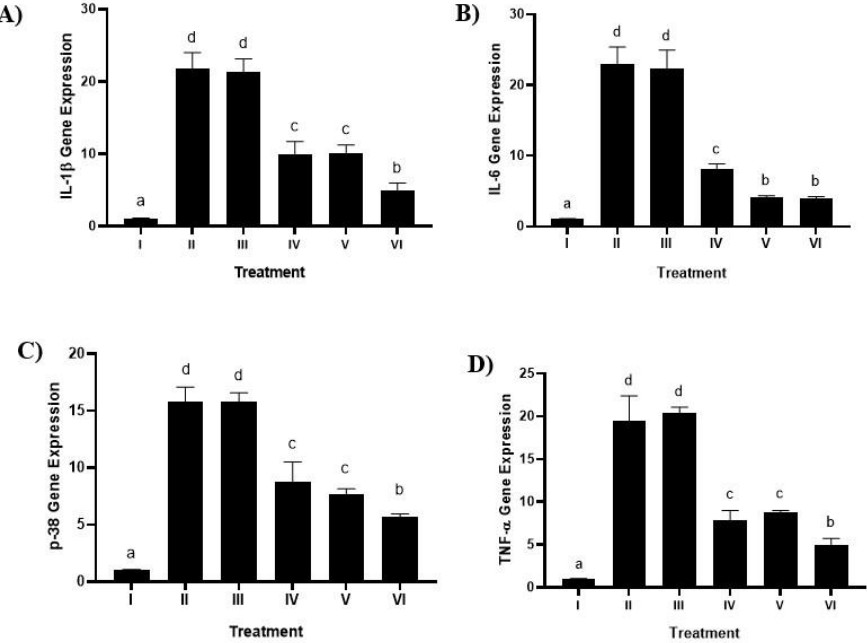

**Figure 7 Effect of brazilin cream toward gene expression in gingivitis rats model.** (A) IL-1 β gene expression (B) IL-6 (C) P-38 dan D) TNF-α. Data are presented as mean ± standard deviation. For each treatment, the assay was performed in four repetitions. I: Negative control (normal rat), II: Positive control (bacterial induction rat), III: Vehicle control (1 x oral base cream), IV: Drug control (1 x patent medicine), V: II + 1 x brazilin cream, VI: II + 2 x brazilin cream. Different lowercase letters (a, b, c, d) mark significant differences between treatment groups ($P < 0.05$, one way ANOVA and Dunnet T3 test).

that the emergence of new blood vessels during gingivitis therapy could serve as a sign of gingivitis healing. According to *Smith et al. (2015)*, the development of these blood vessels occurs due to reduced inflammation, which, in turn, supplies oxygen and nutrients to promote tissue regeneration. Another study also reported that the use of pomegranate gel can reduce gum inflammation and promote tissue regeneration (*Somu et al., 2012*).

In research carried out by *Song et al. (2018)*, gingivitis inflammation is associated with some regulatory T cells and CD4+ T cells that produce the bone-resorptive cytokine receptor activator of nuclear factor-κB (NF-κB), which causes alveolar bone resorption. Patients affected by gingivitis and periodontitis will have proinflammatory cytokines like TNF-α and IL-1β in their gingival tissue and gingival crevicular fluid (*Papathanasiou et al., 2020*). When IL-1β and TNF-α are present, gingival fibroblasts will make more IL-6 and p-38, which will help gingivitis get worse (*Gündogar & Uzunkaya et al., 2021*; *Cavalli et al., 2021*). On the other hand, *Stadler et al. (2016)* observed a decrease in IL-1β in patients with periodontitis, which is a sign of healing. However, if the opposite occurs, such as an increase in IL-1β, it will worsen the condition in patients, causing oral mucositis.

We found that using brazilin cream twice could lower NF-κB levels and the activity of genes that cause inflammation (IL-1β, IL-6, p38, TNF-α, and NF-κB) (Figs. 7A, 7B, 7C, 7D). This result is consistent with studies done by *Kircheis et al. (2020)*, who reported

that partial inhibition of NF-κB expression can inhibit highly pro-inflammatory cytokines and chemokines. *Nirmal et al. (2015)* also stated that brazilin can inhibit the primary regulator of pro-inflammatory cytokine production, the NF-κB signaling pathway. The pro-inflammatory cytokines IL-1β, IL-6, and TNF-α can be inhibited by brazilin through the NF-κB signaling pathway. This is further supported by *Lee et al. (2012)* in their research showing that brazilin showed a protective effect against the loss of fibroblast cell viability caused by UVB and significantly blocked ROS in UVB by inhibiting MMP-1/3 expression, which is a side effect of suppressing NF-κB activation.

Gingivitis is characterized by an inflammatory response involving immune cell infiltration, such as neutrophils, triggered by the presence of bacteria and byproducts from their metabolic processes, leading to disruptions in blood flow and increased collagenolytic activity resulting in extracellular matrix degradation, particularly collagen (*Chandran et al., 2021*). Studies have linked brazilin to a decrease in the expression of genes that promote inflammation. Brazilin has several biological effects, including antibacterial and anti-inflammatory properties (*Olanwanit & Rojanakorn, 2019*). According to *Vij et al. (2023)*, brazilin's anti-inflammatory qualities can counteract reactive oxygen species (ROS), which lowers oxidative stress and inflammation. Gingivitis regulates VEGF, one of the numerous growth factors that mediate the whole angiogenesis process. Research has proven that concentrated growth factors and human gingival fibroblasts enhance gingival regeneration through angiogenesis (*Roi et al., 2022*; *Wang & Yang, 2022*). This fits with what *Ghosh & Gaba (2013)* wrote about how the main effects of active compounds in plant extracts can help with healing and reducing inflammation because they kill germs, are antioxidants, and have active parts that can boost cell growth, blood vessel formation, and collagen production. Apart from that, there are limitations in this study, specifically the limited number of samples and the length of the process, which sometimes fail to fully capture the expected disease development, making it challenging to measure gingival inflammation.

## CONCLUSIONS

Brazilin cream treatment holds promise for gingival tissue repair by mitigating inflammation, promoting angiogenesis, a positive indicator of improved blood vessel formation, and subsequently boosting collagen activity. In addition, it helps lower pro-inflammatory cytokines such as IL-1β, IL-6, p-38, TNF-α, and NF-κB, indicating that the brazilin cream application is effective.

## ACKNOWLEDGEMENTS

We are thankful to the iRATco Veterinary Laboratory Services in Bogor, West Java, Indonesia, and AMUBBRC in Bandung, West Java, Indonesia, for providing laboratory facilities.

### Funding

This work was supported by the Maranatha Christian University which provided fund for this research (Skema Percepatan Guru Besar) with research grant Number 007/PRJ-GB/UKM/I/2023. The funders had no role in study design, data collection and analysis, decision to publish, or preparation of the manuscript.

### Grant Disclosures

The following grant information was disclosed by the authors:
The Maranatha Christian University which provided fund for this research (Skema Percepatan Guru Besar): 007/PRJ-GB/UKM/I/2023.

### Competing Interests

The authors declare there are no competing interests.

### Author Contributions

- Vinna Kurniawati Sugiaman conceived and designed the experiments, analyzed the data, authored or reviewed drafts of the article, and approved the final draft.
- Jeffrey Jeffrey conceived and designed the experiments, analyzed the data, authored or reviewed drafts of the article, and approved the final draft.
- Silvia Naliani conceived and designed the experiments, performed the experiments, prepared figures and/or tables, and approved the final draft.
- Natallia Pranata conceived and designed the experiments, authored or reviewed drafts of the article, and approved the final draft.
- Shelly Lelyana conceived and designed the experiments, performed the experiments, authored or reviewed drafts of the article, and approved the final draft.
- Wahyu Widowati conceived and designed the experiments, analyzed the data, prepared figures and/or tables, authored or reviewed drafts of the article, and approved the final draft.
- Rival Ferdiansyah performed the experiments, analyzed the data, prepared figures and/or tables, and approved the final draft.
- Dhanar Septyawan Hadiprasetyo performed the experiments, prepared figures and/or tables, and approved the final draft.
- Vini Ayuni performed the experiments, prepared figures and/or tables, and approved the final draft.

### Ethics

The following information was supplied relating to ethical approvals (i.e., approving body and any reference numbers):

Animals Ethics Committee of iRATco VLS.

### Data Availability

Measurements of inflammatory gingivitis are available in Fig. 1. The data showed changes in the gingival tissue that became wounds. The injured tissue is used for histopathological, angiogeneic, inflammatory and collagen analysis.

## Supplemental Information

Supplemental information for this article can be found online at http://dx.doi.org/10.7717/peerj.17642#supplemental-information.

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
