# Peer review of "Brazilin cream from Caesalpinia sappan inhibit periodontal disease: in vivo study"

_PeerJ, doi:10.7717/peerj.17642_

## Round 0.1 · original submission · Major Revisions

· Academic Editor

Major Revisions

Dear authors

I would like to express my sincere appreciation for the dedication and effort you have put into conducting your recent study. I am writing to formally request a major review for your manuscript titled (Brazilin Cream from Caesalpinia sappan Inhibit Periodontal disease: In Vivo Study) according to the considerations of the three external peer reviewers.

Firstly, the title of the manuscript appears to be quite generic. I recommend considering a more informative title that accurately reflects the content and focus of your study.

Additionally, it would be beneficial to include the hypothesis of the study in the introduction and discuss the acceptance or rejection of these hypotheses in the discussion section. This would provide clarity regarding the objectives and expectations of the study.

Furthermore, I think that there is a need to address certain potential negative points in the discussion. These include the limited evidence from human trials, the sole focus on Brazilin, the need for a more comprehensive understanding of inflammatory mechanisms, the scope of evaluation, and the generalization from animal models.

Addressing these points will enhance the overall quality and credibility of your manuscript, providing a more balanced evaluation of the potential benefits and limitations of using Brazilin cream for treating periodontitis.

I have included more detailed comments provided by the external peer reviewers below for your reference. I look forward to seeing the revisions made to your manuscript and believe that they will greatly strengthen the impact of your study.

Thank you for your attention to these matters, and I eagerly await the updated version of your manuscript.

**Language Note:** The review process has identified that the English language must be improved. PeerJ can provide language editing services - please contact us at [email protected] for pricing (be sure to provide your manuscript number and title). Alternatively, you should make your own arrangements to improve the language quality and provide details in your response letter. – PeerJ Staff

Reviewer 1 ·

Basic reporting

Congratulations to the authors for their commendable work and for exploring an innovative topic in their study. I would like to suggest a thorough orthographic check of the document, as I have noticed some errors, especially in lines 59 and 60, among others. The readability is satisfactory for the audience.

Experimental design

I recommend a more detailed description of the methodology for reproducibility, especially with regard to sample preparation. Include factors such as room temperature and the drying process mentioned in lines 121 and 122, respectively.
In the decalcification process described in lines 168 to 169, include details of the acid solution used, including concentration and brand. This will improve clarity and reproducibility.
Explain the DNA synthesis from lines 202 to 203 for a more complete understanding.

Validity of the findings

I recommend reviewing line 237, where the term "PC" appears in Figure 2B without prior explanation. I believe this is an error as the figure displays "bc."
In line 246, the term "MT" is used without prior explanation. I assume it stands for "Masson trichome."
In Table 3, I suggest standardizing the images for a clearer comparison, especially for study groups of greater interest (V and VI), which exhibit a less optimal image quality.

Additional comments

I advise the authors to review both the language and sentence structure. I believe this topic is intriguing; however, the article requires enhancement in terms of content, data, and images.
Key points:
1. The methods need more details and better explanations
2. The images could benefit from higher resolution
3. The text needs revision for structure, content, and English language usage

Reviewer 2 ·

Basic reporting

After every full stop in a sentence, the next word must begin in capital letters. In the methods section, future and present tenses are observed. These are glaring errors seen throughout the text and suggests that it was not properly revised by the authors. I suggest you have a colleague who is proficient in English and familiar with the subject matter review your manuscript, or contact a professional editing service.


Line 32-34: This is not the proper definition of periodontal disease. The description of gingivitis implies a simplistic view of its pathogenesis. Please explain why did you choose to use this definition instead of the American Academy of Periodontology’s or alter this sentence to better suit the multifactorial façade of the periodontal disease.
Line 62-63: “Gingivitis can develop into periodontitis, a condition marked by the deterioration of tissues supporting the teeth, if treatment is not received.” Please, provide reference such as the Loe et al., 1986 (DOI: 10.1111/j.1600-051x.1986.tb01487.x) or another credible source.
Line 72-74: There is a difference amongst “risk factors”, “risk indicator” and “disease modifiers”. In these lines the authors mixed the three of them, I suggest further search or remove the indicators and modifiers from the sentence.
Line 51: Brazilin should be in lower case
Line 53: All keywords used must come from MESH (https://www.ncbi.nlm.nih.gov/mesh/) in order to facilitate your manuscript to be found in indexed research tools, such as pubmed, scielo, etc. Therefore: “Anti-inflamation” not found on MESH, should be replace with “Anti-Inflammatory Agent”; “Periodotal” not found on mesh, should be replace with “Periodontal”. However, I strongly suggest that the authors search for more specific keywords.
Line 59: Misspelling “respone”.
Line 88: I believe a “.” shouldn’t exist after “genome”.
118-171: The English language should be improved to ensure that an international audience can clearly understand your text. The current phrasing makes comprehension difficult.
135, 185, 188: correct “60oC” to “60 oC” and so on
151: correct reference
154: “water ad libitum”, Latin writing must be in italic
155-158: in this sentence the authors provided a group nomenclature that differs from the one used in the tables, the reading community will not be able to tell which group the authors are analyzing in the remaining text. Therefore, author must rewrite this paragraph, first stating the name of the group (Group I, II, II, IV, V, VI) and then explain what treatment was performed in each group.
158: “The drug used in the drug group is Kenalog Triacinolone Acetonide ointment 0,1%”. Why did you choose this drug as your positive control? In order to support your choice, please provide high quality literature references where other research groups proved that this medication helps control periodontitis. I understand from your results that it somewhat helped control inflammation, however, in order to be considered a positive control, it needs to be a well-established “gold standard” medication with known periodontitis control effects (which this reviewer is unaware of anything like that).
175-178: you use two different spellings “10 minutes and ten minutes”, please standardize all measurements, since this misspelling is seen in other parts of your manuscript.
176: correct “rinse rith distilled”
183: besides the presence of unsatisfactory conjugation of verb tenses and measurements, the rest of the methods section should follow this writing pattern.
209: Should be “statistical analysis”
216: Figure 1 is a histologic tissue collage; I believe you meant table 3.
234: reference which score graph you are talking about.
262-263: “Inflammation that occurs due to a lack of antioxidants results in an excessive inflammatory response.”. Kindly provide reference
302: “which causes dental bone resorption” I believe you meant alveolar bone
304: correct reference
305: correct misspellings
306-308: “Conversely, Stadler et al. (2016) observed a drop in IL-1 beta periodontitis patients following theraphy is indicating signs of healing.”. Please check the English, I understood what you meant but reading community may not.
311: P38 is spelled different from the rest of the manuscript

Experimental design

Every chemical, equipment, tool, supply used in this study must have the model and maker in parentheses “()” after the first time it is mentioned.
112-114: The study’s aim must be better described in order to make the manuscript more appealing to the reading community
145: what animal facility? Please specify how many rats were used for each part of the study
146-149: Please, better specify your inclusion and non-inclusion criteria (what disease?)
164: “For subjects in this group”, which group?
166: “After 14 days”, 14 days from the beginning of gingivitis induction or from when the rats started showing gingivitis signs?
233-237: Your results are not in accord with your graphs, for example, Group IV had the same results and weren’t mentioned. If you can prove the drug used in Group IV is the gold standard, similarities are a good sign. You can extract more results from your data, the fact that the angiogenesis score is similar to negative control group (Group I) is a good discussion point.
246-250: according to your data at Fig 6, none of the statements you made is correct. Group I showed significant difference with Group II and III (as it should), all the other Groups were similar.
268-271: “The quality of the cream produced meets standards with a homogeneous texture (not lumpy and the particles are evenly distributed), the resulting spreadability is 4.89 with a pH of 8 and the resulting viscosity is 208,000 (Vinna et al., 2024 under review).” This belongs in the results section and the methodology of how the pH, viscosity and spread were measured should be described in the methods section.
180: “The parameters measured were the area of inflammation, angiogenesis, and collagen tissue area”, kindly provide better description of how these measurements were performed in this section

Validity of the findings

Line 210: You mentioned “Qualitative comparison of IHC” on line 195, which statistical analysis did you use for the qualitative comparison? Which test did you used to determine normality?
214: In this section, you must be clear with the community (in the methods section) that it was only a observational statement, that no analysis were made. Please, preferably use the Group names (I, II, II, …) to refer to your groups for the entirety of the manuscript, to avoid confusion.
215-224: “…that could treat the condition of the gums or gingival tissue in rats, although plaque remained but did not cause tartar (Figure 1). This shows that brazilin cream treatment has an antibacterial effect on the cause of gingivitis and can be effective in the treatment of the condition of the gums and teeth.”. This whole statement in quotation has no foundation, first, you did an observational “analysis”, therefore no conclusion can be made from that, only descriptive statements (which you did before the part that it is quoted). Second, it is known that the main cause of biofilm-induced gingivitis is the biofilm, not the calculus (Listgarten & Ellegaard, 1973; Allen & Kerr, 1965; Nyman et al., 1988; Mombelli et al., 1995). Third, you can’t state that it has a microbiological effect if no microbiological analysis (e.g., UFC counts) were performed.
231-233: “Based on the results of histopatholgy of gingival tissues at 40x magnification of the field of view, the number of cells 233 undergoing inflammation and angiogenesis is calculated and then averaged.” This is a part of your methods section, not results.
282-290: this paragraph doesn’t belong in a discussion section
Add a paragraph at the end of the discussion stating all the limitations of the methods used and the one intrinsic to an animal model study
Conclusion
After the correction suggested by all the reviewers, re-elaborate your conclusion.

Additional comments

Dear Authors, after thorough analysis of your manuscript, with the aim of enhancing its overall quality, I kindly request that you meticulously review the following notes and proceed to implement the necessary revisions accordingly.
I strongly suggest you alter the title to gingivitis instead of Periodontal Disease. According to the 2017 Workshop from the American Academy of Periodontology and the European Federation of Periodontology, Gingivitis - which was the condition you actually analyzed – is classified as Gingivitis and not Periodontal Disease. This may mislead the international community when first reading your manuscript.
Therefore, all of your conclusions, discussion must be based on the fact that you analyzed gingivitis, you can’t extend that to the Periodontal Disease.

Annotated reviews are not available for download in order to protect the identity of reviewers who chose to remain anonymous.

·

Basic reporting

The aim of the study is very interesting, but some points need to be clarified and improved. The suggestions are highlighted and commented in the PDF. Please, click on the highlights to see the comments.


The text need some language revision.

Experimental design

My biggest concern is the lack of some important information about the design of the study.

Even though the authors have presented the ARRIVE checklist file, some information given in this checklist are not presented or cannot be found in the text. The authors also did not mention in the text if the study followed this guidelines.

There is no mention of the sample size calculation, how many animals were included, how many animals were analysed, etc.

The induction of gingivitis/periodontitis must be clarifyed.

Validity of the findings

The conclusion of the study is not related to the aim of the study. It must be rewritten.

The authors did not mention the limitations of the study.

Additional comments

No comment.

---

## Round 0.2 · Minor Revisions

· Academic Editor

Minor Revisions

Dear authors,

Based on the feedback from the external peer reviewer, we have determined that the article requires copyediting to address some typographical errors and English grammar issues. Please pay attention to this point to avoid further cycles of reviewing.

I suggest to read your manuscript, without track-changes mode, or having a colleague review it to catch any errors you might have missed.

Reviewer 2 ·

Basic reporting

I highly recommend a professional english translator to grammar check the paper, numerous grammar, verbal tenses, citations and punctuation mistakes can be found in the text, even after this reviewer's previous correction.

The solicited references were provided, now being sufficient.

Figures and raw data are relevant and sufficient.

Experimental design

Although study's aims description have improved, is still not a well defined "research question".

Methods section could also benefit from a professional english grammar check, although it now possesses the necessary parts and descriptions, the incorrect phrasing and verbal tenses will affect the comprehension of the community reading the manuscript.

The p in (p<0.05) is always on lower case, please correct that.

Validity of the findings

no comment

Additional comments

no comment

·

Basic reporting

The authors have improved the quality of the manuscript, providing important additional information that was missing.

Experimental design

The authors have improved the quality of the manuscript, providing important additional information that was missing.

Validity of the findings

The authors have improved the quality of the manuscript, providing important additional information that was missing.

---

## Round 0.3 · accepted · Accept

· Academic Editor

Accept

The article has now been accepted.

Reviewer 2 ·

Basic reporting

no comment

Experimental design

no comment

Validity of the findings

no comment

Additional comments

no comment